# Finite Sample Prediction and Recovery Bounds for Ordinal Embedding

**Lalit Jain**
University of Michigan
Ann Arbor, MI 48109
lalitj@umich.edu

**Kevin Jamieson**
University of California, Berkeley
Berkeley, CA 94720
kjamieson@berkeley.edu

**Robert Nowak**
University of Wisconsin
Madison, WI 53706
rdnowak@wisc.edu

## Abstract

The goal of ordinal embedding is to represent items as points in a low-dimensional Euclidean space given a set of constraints like "item $i$ is closer to item $j$ than item $k$". Ordinal constraints like this often come from human judgments. The classic approach to solving this problem is known as *non-metric multidimensional scaling*. To account for errors and variation in judgments, we consider the noisy situation in which the given constraints are independently corrupted by reversing the correct constraint with some probability. The ordinal embedding problem has been studied for decades, but most past work pays little attention to the question of whether accurate embedding is possible, apart from empirical studies. This paper shows that under a generative data model it is possible to learn the correct embedding from noisy distance comparisons. In establishing this fundamental result, the paper makes several new contributions. First, we derive prediction error bounds for embedding from noisy distance comparisons by exploiting the fact that the rank of a distance matrix of points in $\mathbb{R}^d$ is at most $d + 2$. These bounds characterize how well a learned embedding predicts new comparative judgments. Second, we show that the underlying embedding can be recovered by solving a simple convex optimization. This result is highly non-trivial since we show that the linear map corresponding to distance comparisons is non-invertible, but there exists a nonlinear map that is invertible. Third, two new algorithms for ordinal embedding are proposed and evaluated in experiments.

## 1 Ordinal Embedding

Ordinal embedding aims to represent items as points in $\mathbb{R}^d$ so that the distances between items agree as well as possible with a given set of ordinal comparisons such as item $i$ is closer to item $j$ than to item $k$. In other words, the goal is to find a geometric representation of data that is faithful to comparative similarity judgments. This problem has been studied and applied for more than $50$ years, dating back to the classic *non-metric multidimensional scaling* (NMDS) [1, 2] approach, and it is widely used to gauge and visualize how people perceive similarities.

Despite the widespread application of NMDS and recent algorithmic developments [3, 4, 5, 6, 7], the fundamental question of whether an embedding can be learned from noisy distance/similarity comparisons had not been answered. This paper shows that if the data are generated according to a known probabilistic model, then accurate recovery of the underlying embedding is possible by solving a simple convex optimization, settling this long-standing open question. In the process of answering this question, the paper also characterizes how well a learned embedding predicts new distance comparisons and presents two new computationally efficient algorithms for solving the optimization problem.

## 1.1 Related Work

The classic approach to ordinal embedding is NMDS [1, 2]. Recently, several authors have proposed new approaches based on more modern techniques. *Generalized NMDS* [3] and *Stochastic Triplet Embedding (STE)* [6] employ hinge or logistic loss measures and convex relaxations of the low-dimensionality (i.e., rank) constraint based on the nuclear norm. These works are most closely related to the theory and methods in this paper. The *Linear partial order embedding (LPOE)* method is similar, but starts with a known Euclidean embedding and learns a kernel/metric in this space based distance comparison data [7]. The *Crowd Kernel* [4] and *t-STE* [6] propose alternative non-convex loss measures based on probabilistic generative models. The main contributions in these papers are new optimization methods and experimental studies, but did not address the fundamental question of whether an embedding can be recovered under an assumed generative model. Other recent work has looked at the asymptotics of ordinal embedding, showing that embeddings can be learned as the number of items grows and the items densely populate the embedding space [8, 9, 10]. In contrast, this paper focuses on the practical setting involving a finite set items. Finally, it is known that at least $2dn \log n$ distance comparisons are necessary to learn an embedding of $n$ points in $\mathbb{R}^d$ [5].

## 1.2 Ordinal Embedding from Noisy Data

Consider $n$ points $\boldsymbol{x}_1, \boldsymbol{x}_2, \ldots, \boldsymbol{x}_n \in \mathbb{R}^d$. Let $\boldsymbol{X} = [\boldsymbol{x}_1 \cdots \boldsymbol{x}_n] \in \mathbb{R}^{d \times n}$. The *Euclidean distance matrix* $\boldsymbol{D}^\star$ is defined to have elements $D_{ij}^\star = \|\boldsymbol{x}_i - \boldsymbol{x}_j\|_2^2$. Ordinal embedding is the problem of recovering $\boldsymbol{X}$ given ordinal constraints on distances. This paper focuses on "triplet" constraints of the form $D_{ij}^\star < D_{ik}^\star$, where $1 \leq i \neq j \neq k \leq n$. Furthermore, we only observe noisy indications of these constraints, as follows. Each triplet $t = (i, j, k)$ has an associated probability $p_t$ satisfying

$$p_t > 1/2 \iff \|\boldsymbol{x}_i - \boldsymbol{x}_j\|^2 < \|\boldsymbol{x}_i - \boldsymbol{x}_k\|^2 .$$

Let $\mathcal{S}$ denote a collection of triplets drawn independently and uniformly at random. And for each $t \in \mathcal{S}$ we observe an independent random variable $y_t = -1$ with probability $p_t$, and $y_t = 1$ otherwise. The goal is to recover the embedding $\boldsymbol{X}$ from these data. Exact recovery of $\boldsymbol{D}^\star$ from such data requires a known *link* between $p_t$ and $\boldsymbol{D}^\star$. To this end, our main focus is the following problem.

---

**Ordinal Embedding from Noisy Data**

Consider $n$ points $\boldsymbol{x}_1, \boldsymbol{x}_2 \cdots, \boldsymbol{x}_n$ in $d$-dimensional Euclidean space. Let $\mathcal{S}$ denote a collection of triplets and for each $t \in \mathcal{S}$ observe an independent random variable

$$y_t = \begin{cases} -1 & w.p. \ f(D_{ij}^\star - D_{ik}^\star) \\ 1 & w.p. \ 1 - f(D_{ij}^\star - D_{ik}^\star) \end{cases} .$$

where the link function $f : \mathbb{R} \to [0, 1]$ is known. Estimate $\boldsymbol{X}$ from $\mathcal{S}$, $\{y_t\}$, and $f$.

---

For example, if $f$ is the logistic function, then for triplet $t = (i, j, k)$

$$p_t = \mathbb{P}(y_t = -1) = f(D_{ij}^\star - D_{ik}^\star) = \frac{1}{1 + \exp(D_{ij}^\star - D_{ik}^\star)} , \tag{1}$$

then $D_{ij}^\star - D_{ik}^\star = \log\left(\frac{1 - p_t}{p_t}\right)$. However, we stress that we only require the existence of a link function for exact recovery of $\boldsymbol{D}^\star$. Indeed, if one just wishes to *predict* the answers to unobserved triplets, then the results of Section 2 hold for arbitrary $p_t$ probabilities. Aspects of the statistical analysis are related to one-bit matrix completion and rank aggregation [11, 12, 13]. However, we use novel methods for the recovery of the embedding based on geometric properties of Euclidean distance matrices.

## 1.3 Organization of Paper

This paper takes the following approach to ordinal embedding.

1. Our samples are assumed to be independently generated according to a probabilistic model based on an underlying low-rank distance matrix. We use relatively standard statistically learning theory

techniques to analyze the minimizer of a bounded, Lipschitz loss with a nuclear norm constraint, and show that an embedding can be learned from the data that predicts nearly as well as the true embedding with $O(dn \log n)$ samples (Theorem 1).

2. Next, assuming the form of the probabilistic generative model is known (e.g., logistic), we show that if the learned embedding is a good predictor of the ordinal comparisons, then it must also be a good estimator of the true differences of distances between the embedding points (Theorem 2). This result hinges on the fact that the (linear) observation model acts approximately like an isometry on differences of distances.

3. While the true differences of distances can be estimated, the observation process is "blind" to the mean distance between embedding points. Despite this, we show that the mean is determined by the differences of distances, due to the special properties of Euclidean distance matrices. Specifically, the second eigenvalue of the "mean-centered" distance matrix (well-estimated by the data from the estimate of the differences of distances, Theorem 3) is proportional to the mean distance (Theorem 4). This allows us to show that the minimizer of the loss with a nuclear norm constraint indeed recovers an accurate estimate of the underlying true distance matrix.

### 1.4 Notation and Assumptions

We will use $(\boldsymbol{D}^\star, \boldsymbol{G}^\star)$ to denote the distance and Gram matrices of the latent embedding, and $(\boldsymbol{D}, \boldsymbol{G})$ to denote an arbitrary distance matrix and its corresponding Gram matrix. The observations $\{y_t\}$ carry information about $\boldsymbol{D}^\star$, but distance matrices are invariant to rotation and translation, and therefore it may only be possible to recover $\boldsymbol{X}$ up to a rigid transformation. Without loss of generality, we assume assume the points $\boldsymbol{x}_1, \ldots \boldsymbol{x}_n \in \mathbb{R}^d$ are centered at the origin (i.e., $\sum_{i=1}^n \boldsymbol{x}_i = \boldsymbol{0}$).

Define the *centering matrix* $\boldsymbol{V} := \boldsymbol{I} - \frac{1}{n}\boldsymbol{1}\boldsymbol{1}^T$. If $\boldsymbol{X}$ is centered, $\boldsymbol{X}\boldsymbol{V} = \boldsymbol{X}$. Note that $\boldsymbol{D}^\star$ is determined by the Gram matrix $\boldsymbol{G}^\star = \boldsymbol{X}^T\boldsymbol{X}$. In addition, $\boldsymbol{X}$ can be determined from $\boldsymbol{G}^\star$ up to a unitary transformation. Note that if $\boldsymbol{X}$ is centered, the Gram matrix is "centered" so that $\boldsymbol{V}\boldsymbol{G}^\star\boldsymbol{V} = \boldsymbol{G}^\star$. It will be convenient in the paper to work with both the distance and Gram matrix representations, and the following identities will be useful to keep in mind. For any distance matrix $\boldsymbol{D}$ and its centered Gram matrix $\boldsymbol{G}$

$$\boldsymbol{G} = -\frac{1}{2}\boldsymbol{V}\boldsymbol{D}\boldsymbol{V} , \tag{2}$$

$$\boldsymbol{D} = \operatorname{diag}(\boldsymbol{G})\boldsymbol{1}^T - 2\boldsymbol{G} + \boldsymbol{1}\operatorname{diag}(\boldsymbol{G})^T , \tag{3}$$

where $\operatorname{diag}(\boldsymbol{G})$ is the column vector composed of the diagonal of $\boldsymbol{G}$. In particular this establishes a bijection between centered Gram matrices and distance matrices. We refer the reader to [14] for an insightful and thorough treatment of the properties of distance matrices. We also define the set of all unique triplets

$$\mathcal{T} := \big\{(i,j,k) : 1 \leq i \neq j \neq k \leq n, j < k\big\} .$$

**Assumption 1.** *The observed triplets in $\mathcal{S}$ are drawn independently and unifomly from $\mathcal{T}$.*

## 2 Prediction Error Bounds

For $t \in \mathcal{T}$ with $t = (i,j,k)$ we define $\mathcal{L}_t$ to be the linear operator satisfying $\mathcal{L}_t(\boldsymbol{X}^T\boldsymbol{X}) = \|\boldsymbol{x}_i - \boldsymbol{x}_j\|^2 - \|\boldsymbol{x}_i - \boldsymbol{x}_k\|^2$ for all $t \in \mathcal{T}$. In general, for any Gram matrix $\boldsymbol{G}$

$$\mathcal{L}_t(\boldsymbol{G}) := G_{jj} - 2G_{ij} - G_{kk} + 2G_{ik}.$$

We can naturally view $\mathcal{L}_t$ as a linear operator on $\mathbb{S}_+^n$, the space of $n \times n$ symmetric positive semidefinite matrices. We can also represent $\mathcal{L}_t$ as a symmetric $n \times n$ matrix that is zero everywhere except on the submatrix corresponding to $i, j, k$ which has the form

$$\begin{bmatrix} 0 & -1 & 1 \\ -1 & 1 & 0 \\ 1 & 0 & -1 \end{bmatrix}$$

and so we will write

$$\mathcal{L}_t(\boldsymbol{G}) := \langle \mathcal{L}_t, \boldsymbol{G} \rangle$$

where $\langle A, B \rangle = \text{vec}(A)^T \text{vec}(B)$ for any compatible matrices $A, B$. Ordering the elements of $\mathcal{T}$ lexicographically, we arrange all the $\mathcal{L}_t(\boldsymbol{G})$ together to define the $n\binom{n-1}{2}$-dimensional vector

$$\mathcal{L}(\boldsymbol{G}) = [\mathcal{L}_{123}(\boldsymbol{G}), \mathcal{L}_{124}(\boldsymbol{G}), \cdots, \mathcal{L}_{ijk}(\boldsymbol{G}), \cdots]^T. \tag{4}$$

Let $\ell(y_t \langle \mathcal{L}_t, \boldsymbol{G} \rangle)$ denote a loss function. For example we can consider the $0 - 1$ loss $\ell(y_t \langle \mathcal{L}_t, \boldsymbol{G} \rangle) = \mathbb{1}_{\{\text{sign}\{y_t \langle \mathcal{L}_t, \boldsymbol{G} \rangle\} \neq 1\}}$, the hinge-loss $\ell(y_t \langle \mathcal{L}_t, \boldsymbol{G} \rangle) = \max\{0, 1 - y_t \langle \mathcal{L}_t, \boldsymbol{G} \rangle\}$, or the logistic loss

$$\ell(y_t \langle \mathcal{L}_t, \boldsymbol{G} \rangle) = \log(1 + \exp(-y_t \langle \mathcal{L}_t, \boldsymbol{G} \rangle)). \tag{5}$$

Let $p_t := \mathbb{P}(y_t = -1)$ and take the expectation of the loss with respect to both the uniformly random selection of the triple $t$ and the observation $y_t$, we have the risk of $\boldsymbol{G}$

$$R(\boldsymbol{G}) \quad := \quad \mathbb{E}[\ell(y_t \langle \mathcal{L}_t, \boldsymbol{G} \rangle)] = \frac{1}{|\mathcal{T}|} \sum_{t \in \mathcal{T}} p_t \ell(-\langle \mathcal{L}_t, \boldsymbol{G} \rangle) + (1 - p_t)\ell(\langle \mathcal{L}_t, \boldsymbol{G} \rangle).$$

Given a set of observations $\mathcal{S}$ under the model defined in the problem statement, the empirical risk is,

$$\widehat{R}_{\mathcal{S}}(\boldsymbol{G}) = \frac{1}{|\mathcal{S}|} \sum_{t \in \mathcal{S}} \ell(y_t \langle \mathcal{L}_t, \boldsymbol{G} \rangle) \tag{6}$$

which is an unbiased estimator of the true risk: $\mathbb{E}[\widehat{R}_{\mathcal{S}}(\boldsymbol{G})] = R(\boldsymbol{G})$. For any $\boldsymbol{G} \in \mathbb{S}_+^n$, let $\|\boldsymbol{G}\|_*$ denote the nuclear norm and $\|\boldsymbol{G}\|_\infty := \max_{ij} |\boldsymbol{G}_{ij}|$. Define the constraint set

$$\mathcal{G}_{\lambda,\gamma} := \{\boldsymbol{G} \in \mathbb{S}_+^n : \|\boldsymbol{G}\|_* \leq \lambda, \|\boldsymbol{G}\|_\infty \leq \gamma\}. \tag{7}$$

We estimate $\boldsymbol{G}^\star$ by $\widehat{\boldsymbol{G}}$, the solution of the program,

$$\widehat{\boldsymbol{G}} := \underset{\boldsymbol{G} \in \mathcal{G}_{\lambda,\gamma}}{\text{argmin}} \widehat{R}_{\mathcal{S}}(\boldsymbol{G}). \tag{8}$$

Since $\boldsymbol{G}^\star$ is positive semidefinite, we expect the diagonal entries of $\boldsymbol{G}^\star$ to bound the off-diagonal entries. So an infinity norm constraint on the diagonal guarantees that the points $\boldsymbol{x}_1, \ldots, \boldsymbol{x}_n$ corresponding to $\boldsymbol{G}^\star$ live inside a bounded $\ell_2$ ball. The $\ell_\infty$ constraint in (7) plays two roles: 1) if our loss function is Lipschitz, large magnitude values of $\langle \mathcal{L}_t, \boldsymbol{G} \rangle$ can lead to large deviations of $\widehat{R}_{\mathcal{S}}(\boldsymbol{G})$ from $R(\boldsymbol{G})$; bounding $\|\boldsymbol{G}\|_\infty$ bounds $|\langle \mathcal{L}_t, \boldsymbol{G} \rangle|$. 2) Later we will define $\ell$ in terms of the link function $f$ and as the magnitude of $\langle \mathcal{L}_t, \boldsymbol{G} \rangle$ increases the magnitude of the derivative of the link function $f$ typically becomes very small, making it difficult to "invert"; bounding $\|\boldsymbol{G}\|_\infty$ tends to keep $\langle \mathcal{L}_t, \boldsymbol{G} \rangle$ within an invertible regime of $f$.

**Theorem 1.** *Fix $\lambda, \gamma$ and assume $\boldsymbol{G}^\star \in \mathcal{G}_{\lambda,\gamma}$. If the loss function $\ell(\cdot)$ is $L$-Lipschitz (or $|\sup_y \ell(y)| \leq L \max\{1, 12\gamma\}$) then with probability at least $1 - \delta$,*

$$R(\widehat{\boldsymbol{G}}) - R(\boldsymbol{G}^\star) \leq \frac{4L\lambda}{|\mathcal{S}|} \left( \sqrt{\frac{18|\mathcal{S}|\log(n)}{n}} + \frac{\sqrt{3}}{3}\log n \right) + L\gamma\sqrt{\frac{288\log 2/\delta}{|\mathcal{S}|}}$$

*Proof.* The proof follows from standard statistical learning theory techniques, see for instance [15]. By the bounded difference inequality, with probability $1 - \delta$

$$R(\widehat{\boldsymbol{G}}) - R(\boldsymbol{G}^\star) = R(\widehat{\boldsymbol{G}}) - \widehat{R}_{\mathcal{S}}(\widehat{\boldsymbol{G}}) + \widehat{R}_{\mathcal{S}}(\widehat{\boldsymbol{G}}) - \widehat{R}_{\mathcal{S}}(\boldsymbol{G}^\star) + \widehat{R}_{\mathcal{S}}(\boldsymbol{G}^\star) - R(\boldsymbol{G}^\star)$$

$$\leq 2 \sup_{\boldsymbol{G} \in \mathcal{G}_{\lambda,\gamma}} |\widehat{R}_{\mathcal{S}}(\boldsymbol{G}) - R(\boldsymbol{G})| \leq 2\mathbb{E}[\sup_{\boldsymbol{G} \in \mathcal{G}_{\lambda,\gamma}} |\widehat{R}_{\mathcal{S}}(\boldsymbol{G}) - R(\boldsymbol{G})|] + \sqrt{\frac{2B^2 \log 2/\delta}{|S|}}$$

where $\sup_{\boldsymbol{G} \in \mathcal{G}_{\lambda,\gamma}} \ell(y_t \langle \mathcal{L}_t, \boldsymbol{G} \rangle) - \ell(y_{t'} \langle \mathcal{L}_{t'}, \boldsymbol{G} \rangle) \leq \sup_{\boldsymbol{G} \in \mathcal{G}_{\lambda,\gamma}} L|\langle y_t \mathcal{L}_t - y_{t'} \mathcal{L}_{t'}, \boldsymbol{G} \rangle| \leq 12L\gamma =: B$ using the facts that $\mathcal{L}_t$ has 6 non-zeros of magnitude 1 and $\|\boldsymbol{G}\|_\infty \leq \gamma$.

Using standard symmetrization and contraction lemmas, we can introduce Rademacher random variables $\epsilon_t \in \{-1, 1\}$ for all $t \in \mathcal{S}$ so that

$$\mathbb{E} \sup_{\boldsymbol{G} \in \mathcal{G}_{\lambda,\gamma}} |\widehat{R}_{\mathcal{S}}(\boldsymbol{G}) - R(\boldsymbol{G})| \leq \mathbb{E} \sup_{\boldsymbol{G} \in \mathcal{G}_{\lambda,\gamma}} \frac{2L}{|\mathcal{S}|} \left| \sum_{t \in \mathcal{S}} \epsilon_t \langle \mathcal{L}_t, \boldsymbol{G} \rangle \right|.$$

The right hand side is just the Rademacher complexity of $\mathcal{G}_{\lambda,\gamma}$. By definition,

$$\{\boldsymbol{G} : \|\boldsymbol{G}\|_* \le \lambda\} = \lambda \cdot \mathrm{conv}(\{uu^T : |u| = 1\}).$$

where $\mathrm{conv}(U)$ is the convex hull of a set $U$. Since the Rademacher complexity of a set is the same as the Rademacher complexity of it's closed convex hull,

$$\mathbb{E} \sup_{\boldsymbol{G} \in \mathcal{G}_{\lambda,\gamma}} \left| \sum_{t \in \mathcal{S}} \epsilon_t \langle \mathcal{L}_t, \boldsymbol{G} \rangle \right| \le \lambda \mathbb{E} \sup_{|u|=1} \left| \sum_{t \in \mathcal{S}} \epsilon_t \langle \mathcal{L}_t, uu^T \rangle \right| = \lambda \mathbb{E} \sup_{|u|=1} \left| u^T \left( \sum_{t \in \mathcal{S}} \epsilon_t \mathcal{L}_t \right) u \right|$$

which we recognize is just $\lambda \mathbb{E} \| \sum_{t \in \mathcal{S}} \epsilon_t \mathcal{L}_t \|$. By [16, 6.6.1] we can bound the operator norm $\| \sum_{t \in S} \epsilon_t \mathcal{L}_t \|$ in terms of the variance of $\sum_{t \in \mathcal{S}} \mathcal{L}_t^2$ and the maximal eigenvalue of $\max_t \mathcal{L}_t$. These are computed in Lemma 1 given in the supplemental materials. Combining these results gives,

$$\frac{2L\lambda}{|\mathcal{S}|} \mathbb{E}\| \sum_{t \in \mathcal{S}} \epsilon_t \mathcal{L}_t \| \le \frac{2L\lambda}{|\mathcal{S}|} \left( \sqrt{\frac{18|\mathcal{S}|\log(n)}{n}} + \frac{\sqrt{3}}{3} \log n \right).$$

$\square$

We remark that if $\boldsymbol{G}$ is a rank $d < n$ matrix then

$$\|\boldsymbol{G}\|_* \le \sqrt{d}\|\boldsymbol{G}\|_F \le \sqrt{d}n\|\boldsymbol{G}\|_\infty$$

so if $\boldsymbol{G}^\star$ is low rank, we really only need a bound on the infinity norm of our constraint set. Under the assumption that $\boldsymbol{G}^\star$ is rank $d$ with $\|\boldsymbol{G}^\star\|_\infty \le \gamma$ and we set $\lambda = \sqrt{d}n\gamma$, then Theorem 1 implies that for $|S| > n \log n / 161$

$$R(\widehat{\boldsymbol{G}}) - R(\boldsymbol{G}^\star) \le 8L\gamma \sqrt{\frac{18dn\log(n)}{|\mathcal{S}|}} + L\gamma \sqrt{\frac{288 \log 2/\delta}{|\mathcal{S}|}}$$

with probability at least $1 - \delta$. The above display says that $|\mathcal{S}|$ must scale like $dn \log(n)$ which is consistent with known finite sample bounds [5].

## 3 Maximum Likelihood Embedding

We now turn our attention to recovering metric information about $\boldsymbol{G}^\star$. Let $\mathcal{S}$ be a collection of triplets sampled uniformly at random with replacement and let $f : \mathbb{R} \to (0,1)$ be a known probability function governing the observations. Any link function $f$ induces a natural loss function $\ell_f$, namely, the negative log-likelihood of a solution $\boldsymbol{G}$ given an observation $y_t$ defined as

$$\ell_f(y_t \langle \mathcal{L}_t, \boldsymbol{G} \rangle) = \mathbb{1}_{y_t=-1} \log(\tfrac{1}{f(\langle \mathcal{L}_t, \boldsymbol{G} \rangle)}) + \mathbb{1}_{y_t=1} \log(\tfrac{1}{1-f(\langle \mathcal{L}_t, \boldsymbol{G} \rangle)})$$

For example, the logistic link function of (1) induces the logistic loss of (5). Recalling that $\mathbb{P}(y_t = -1) = f(\langle \mathcal{L}_t, \boldsymbol{G} \rangle)$ we have

$$\mathbb{E}[\ell_f(y_t \langle \mathcal{L}_t, \boldsymbol{G} \rangle)] = f(\langle \mathcal{L}_t, \boldsymbol{G}^\star \rangle) \log(\tfrac{1}{f(\langle \mathcal{L}_t, \boldsymbol{G} \rangle)}) + (1 - f(\langle \mathcal{L}_t, \boldsymbol{G}^\star \rangle)) \log(\tfrac{1}{1-f(\langle \mathcal{L}_t, \boldsymbol{G} \rangle)})$$
$$= H(f(\langle \mathcal{L}_t, \boldsymbol{G}^\star \rangle)) + KL(f(\langle \mathcal{L}_t, \boldsymbol{G}^\star \rangle)|f(\langle \mathcal{L}_t, \boldsymbol{G} \rangle))$$

where $H(p) = p \log(\frac{1}{p}) + (1-p) \log(\frac{1}{1-p})$ and $KL(p,q) = p \log(\frac{p}{q}) + (1-p) \log(\frac{1-p}{1-q})$ are the entropy and KL divergence of Bernoulli RVs with means $p, q$. Recall that $\|\boldsymbol{G}\|_\infty \le \gamma$ controls the magnitude of $\langle \mathcal{L}_t, \boldsymbol{G} \rangle$ so for the moment, assume this is small. Then by a Taylor series $f(\langle \mathcal{L}_t, \boldsymbol{G} \rangle) \approx \frac{1}{2} + f'(0)\langle \mathcal{L}_t, \boldsymbol{G} \rangle$ using the fact that $f(0) = \frac{1}{2}$, and by another Taylor series we have

$$KL(f(\langle \mathcal{L}_t, \boldsymbol{G}^\star \rangle)|f(\langle \mathcal{L}_t, \boldsymbol{G} \rangle)) \approx KL(\tfrac{1}{2} + f'(0)\langle \mathcal{L}_t, \boldsymbol{G}^\star \rangle | \tfrac{1}{2} + f'(0)\langle \mathcal{L}_t, \boldsymbol{G} \rangle)$$
$$\approx 2f'(0)^2(\langle \mathcal{L}_t, \boldsymbol{G}^\star - \boldsymbol{G} \rangle)^2.$$

Thus, recalling the definition of $\mathcal{L}(\boldsymbol{G})$ from (4) we conclude that if $\widetilde{\boldsymbol{G}} \in \arg\min_{\boldsymbol{G}} R(\boldsymbol{G})$ with $R(\boldsymbol{G}) = \frac{1}{|\mathcal{T}|} \sum_{t \in \mathcal{T}} \mathbb{E}[\ell_f(y_t \langle \mathcal{L}_t, \boldsymbol{G} \rangle)]$ then one would expect $\mathcal{L}(\widetilde{\boldsymbol{G}}) \approx \mathcal{L}(\boldsymbol{G}^\star)$. Moreover, since $\widehat{R}_{\mathcal{S}}(\boldsymbol{G})$ is an unbiased estimator of $R(\boldsymbol{G})$, one expects $\mathcal{L}(\widehat{\boldsymbol{G}})$ to approximate $\mathcal{L}(\boldsymbol{G}^\star)$. The next theorem, combined with Theorem 1, formalizes this observation; its proof is found in the appendix.

**Theorem 2.** *Let $C_f = \min_{t \in \mathcal{T}} \inf_{\boldsymbol{G} \in \mathcal{G}_{\lambda,\gamma}} |f'(\langle \mathcal{L}_t, \boldsymbol{G} \rangle)|$ where $f'$ denotes the derivative of $f$. Then for any $\boldsymbol{G}$*

$$\frac{2C_f^2}{|\mathcal{T}|} \|\mathcal{L}(\boldsymbol{G}) - \mathcal{L}(\boldsymbol{G}^\star)\|_F^2 \;\leq\; R(\boldsymbol{G}) - R(\boldsymbol{G}^\star) \,.$$

Note that if $f$ is the logistic link function of (1) then its straightforward to show that $|f'(\langle \mathcal{L}_t, \boldsymbol{G} \rangle)| \geq \frac{1}{4}\exp(-|\langle \mathcal{L}_t, \boldsymbol{G} \rangle|) \geq \frac{1}{4}\exp(-6\|\boldsymbol{G}\|_\infty)$ for any $t$, $\boldsymbol{G}$ so it suffices to take $C_f = \frac{1}{4}\exp(-6\gamma)$.

It remains to see that we can recover $\boldsymbol{G}^\star$ even given $\mathcal{L}(\boldsymbol{G}^\star)$, much less $\mathcal{L}(\widehat{\boldsymbol{G}})$. To do this, it is more convenient to work with distance matrices instead of Gram matrices. Analogous to the operators $\mathcal{L}_t(\boldsymbol{G})$ defined above, we define the operators $\Delta_t$ for $t \in \mathcal{T}$ satisfying,

$$\Delta_t(\boldsymbol{D}) := D_{ij} - D_{ik} \equiv \mathcal{L}_t(\boldsymbol{G}) \,.$$

We will view the $\Delta_t$ as linear operators on the space of symmetric hollow $n \times n$ matrices $\mathbb{S}_h^n$, which includes distance matrices as special cases. As with $\mathcal{L}$, we can arrange all the $\Delta_t$ together, ordering the $t \in \mathcal{T}$ lexicographically, to define the $n\binom{n-1}{2}$-dimensional vector

$$\Delta(\boldsymbol{D}) = [D_{12} - D_{13}, \cdots, D_{ij} - D_{ik}, \cdots]^T.$$

We will use the fact that $\mathcal{L}(\boldsymbol{G}) \equiv \Delta(\boldsymbol{D})$ heavily. Because $\Delta(\boldsymbol{D})$ consists of differences of matrix entries, $\Delta$ has a non-trivial kernel. However, it is easy to see that $\boldsymbol{D}$ can be recovered given $\Delta(\boldsymbol{D})$ *and* any one off-diagonal element of $\boldsymbol{D}$, so the kernel is 1-dimensional. Also, the kernel is easy to identify by example. Consider the regular simplex in $d$ dimensions. The distances between all $n = d+1$ vertices are equal and the distance matrix can easily be seen to be $\boldsymbol{1}\boldsymbol{1}^T - \boldsymbol{I}$. Thus $\Delta(\boldsymbol{D}) = \boldsymbol{0}$ in this case. This gives us the following simple result.

**Lemma 2.** *Let $\mathbb{S}_h^n$ denote the space of symmetric hollow matrices, which includes all distance matrices. For any $\boldsymbol{D} \in \mathbb{S}_h^n$, the set of linear functionals $\{\Delta_t(\boldsymbol{D}), t \in \mathcal{T}\}$ spans an $\binom{n}{2} - 1$ dimensional subspace of $\mathbb{S}_h^n$, and the 1-dimensional kernel is given by the span of $\boldsymbol{1}\boldsymbol{1}^T - \boldsymbol{I}$.*

So we see that the operator $\Delta$ is not invertible on $\mathbb{S}_h^n$. Define $\boldsymbol{J} := \boldsymbol{1}\boldsymbol{1}^T - \boldsymbol{I}$. For any $\boldsymbol{D}$, let $\boldsymbol{C}$, *the centered distance matrix*, be the component of $\boldsymbol{D}$ orthogonal to the kernel of $\mathcal{L}$ (i.e., $\text{tr}(\boldsymbol{C}\boldsymbol{J}) = 0$). Then we have the orthogonal decomposition

$$\boldsymbol{D} \;=\; \boldsymbol{C} \;+\; \sigma_D\, \boldsymbol{J} \,,$$

where $\sigma_D = \text{trace}(\boldsymbol{D}\boldsymbol{J})/\|\boldsymbol{J}\|_F^2$. Since $\boldsymbol{G}$ is assumed to be centered, the value of $\sigma_D$ has a simple interpretation:

$$\sigma_D = \frac{1}{2\binom{n}{2}} \sum_{1 \leq i \leq j \leq n} D_{ij} = \frac{2}{n-1} \sum_{1 \leq i \leq n} \langle x_i, x_i \rangle = \frac{2\|\boldsymbol{G}\|_*}{n-1}, \tag{9}$$

the average of the squared distances or alternatively a scaled version of the nuclear norm of $\boldsymbol{G}$.

Let $\widehat{\boldsymbol{D}}$ and $\widehat{\boldsymbol{C}}$ be the corresponding distance and centered distance matrices corresponding to $\widehat{\boldsymbol{G}}$ the solution to 8. Though $\Delta$ is not invertible on all $\mathbb{S}_h^n$, it is invertible on the subspace orthogonal to the kernel, namely $\boldsymbol{J}^\perp$. So if $\Delta(\widehat{\boldsymbol{D}}) \approx \Delta(\boldsymbol{D}^\star)$, or equivalently $\mathcal{L}(\widehat{\boldsymbol{G}}) \approx \mathcal{L}(\boldsymbol{G}^\star)$, we expect $\widehat{\boldsymbol{C}}$ to be close to $\boldsymbol{C}^\star$. The next theorem quantifies this.

**Theorem 3.** *Consider the setting of Theorems 1 and 2 and let $\widehat{\boldsymbol{C}}, \boldsymbol{C}^\star$ be defined as above. Then*

$$\frac{1}{2\binom{n}{2}} \|\widehat{\boldsymbol{C}} - \boldsymbol{C}^\star\|_F^2 \;\leq\; \frac{L\lambda}{4C_f^2 |\mathcal{S}|}\left(\sqrt{\frac{18|\mathcal{S}|\log(n)}{n}} + \frac{\sqrt{3}}{3}\log n\right) + \frac{L\gamma}{4C_f^2}\sqrt{\frac{288\log 2/\delta}{|\mathcal{S}|}}$$

*Proof.* By combining Theorem 2 with the prediction error bounds obtainined in 1 we see that

$$\frac{2C_f^2}{n\binom{n-1}{2}} \|\mathcal{L}(\widehat{\boldsymbol{G}}) - \mathcal{L}(\boldsymbol{G}^\star)\|_F^2 \leq \frac{4L\lambda}{|\mathcal{S}|}\left(\sqrt{\frac{18|\mathcal{S}|\log(n)}{n}} + \frac{\sqrt{3}}{3}\log n\right) + L\gamma\sqrt{\frac{288\log 2/\delta}{|\mathcal{S}|}}.$$

Next we employ the following *restricted isometry property* of $\Delta$ on the subspace $\boldsymbol{J}^\perp$ whose proof is in the supplementary materials.

**Lemma 3.** *Let $D$ and $D'$ be two different distance matrices of $n$ points in $\mathbb{R}^d$ and $\mathbb{R}^{d'}$. Let $C$ and $C'$ be the components of $D$ and $D'$ orthogonal to $J$. Then*

$$n\|C - C'\|_F^2 \ \leq \ \|\Delta(C) - \Delta(C')\|^2 = \|\Delta(D) - \Delta(D')\|^2 \ \leq \ 2(n-1)\|C - C'\|_F^2 \ .$$

The result then follows. $\qquad\square$

This implies that by collecting enough samples, we can recover the centered distance matrix. By applying the discussion following Theorem 1 when $G^\star$ is rank $d$, we can state an upperbound of $\frac{1}{2\binom{n}{2}}\|\widehat{C} - C^\star\|_F^2 \ \leq O\left(\frac{L\gamma}{C_f^2}\sqrt{\frac{dn\log(n)+\log(1/\delta)}{|\mathcal{S}|}}\right)$. However, it is still not clear that this is enough to recover $D^\star$ or $G^\star$. Remarkably, despite this unknown component being in the kernel, we show next that it can be recovered.

**Theorem 4.** *Let $D$ be a distance matrix of $n$ points in $\mathbb{R}^d$, let $C$ be the component of $D$ orthogonal to the kernel of $\mathcal{L}$, and let $\lambda_2(C)$ denote the second largest eigenvalue of $C$. If $n > d + 2$, then*

$$D \ = \ C \ + \ \lambda_2(C)\,J \ . \tag{10}$$

This shows that $D$ is uniquely determined as a function of $C$. Therefore, since $\Delta(D) = \Delta(C)$ and because $C$ is orthogonal to the kernel of $\Delta$, the distance matrix $D$ can be recovered from $\Delta(D)$, even though the linear operator $\Delta$ is non-invertible.

We now provide a proof of Theorem 4 in the case where $n > d + 3$. The result is true in the case when $n > d + 2$ but requires a more detailed analysis. This includes the construction of a vector $x$ such that $Dx = 1$ and $1^T x \geq 0$ for any distance matrix a result in [17].

*Proof.* To prove Theorem 4 we need the following lemma, proved in the supplementary materials.

**Lemma 4.** *Let $D$ be a Euclidean distance matrix on $n$ points. Then $D$ is negative semidefinite on the subspace*

$$1^\perp := \{x \in \mathbb{R}^n | 1^T x = 0\}.$$

*Furthermore,* $\ker(D) \subset 1^\perp$.

For any matrix $M$, let $\lambda_i(M)$ denote its $i$th largest eigenvalue. Under the conditions of the theorem, we show that for $\sigma > 0$, $\lambda_2(D - \sigma J) = \sigma$. Since $C = D - \sigma_D J$, this proves the theorem.

Note that, $\lambda_i(D - \sigma J) = \lambda_i(D - \sigma 11^T) + \sigma$ for $1 \leq i \leq n$ and $\sigma$ arbitrary. So it suffices to show that $\lambda_2(D - \sigma 11^T) = 0$.

By Weyl's Theorem

$$\lambda_2(D - \sigma 11^T) \ \leq \ \lambda_2(D) + \lambda_1(-\sigma 11^T) \ .$$

Since $\lambda_1(-\sigma 11^T) = 0$, we have $\lambda_2(D - \sigma 11^T) \leq \lambda_2(D) = 0$. By the Courant-Fischer Theorem

$$\lambda_2(D) \ = \ \min_{U:\dim(U)=n-1} \max_{x \in U, x \neq 0} \frac{x^T D x}{x^T x} \ \leq \ \min_{U=1^\perp} \max_{x \in U, x \neq 0} \frac{x^T D x}{x^T x} \ \leq \ 0$$

since $D$ negative semidefinite on $1^\perp$. Now let $v_i$ denote the $i$th eigenvector of $D$ with eigenvalue $\lambda_i = 0$. Then

$$(D - \sigma 11^T)v_i = Dv_i = 0 \ ,$$

since $v_i^T 1 = 0$ by 4. So $D - \sigma 11^T$ has at least $n - d - 2$ zero eigenvalues, since $\text{rank} D \leq d + 2$. In particular, if $n > d + 3$, then $D - \sigma 11^T$ must have at least two eigenvalues equal to $0$. Therefore, $\lambda_2(D - \sigma 11^T) = 0$. $\qquad\square$

The previous theorem along with Theorem 3 guarantees that we can recover $G^\star$ as we increase the number of triplets sampled. The final theorem, which follows directly from Theorems 3 and 4, summarizes this.

**Theorem 5.** *Assume $n > d + 2$ and consider the setting of Theorems 1 and 2. As $|\mathcal{S}| \to \infty$, $\widehat{D} \to D^\star$ where $\widehat{D}$ is the distance matrix corresponding to $\widehat{G}$ (the solution to 8).*

*Proof.* Recall $\widehat{D} = \widehat{C} + \lambda_2(\widehat{C})J$, so as $\widehat{C} \to C^\star$, $\widehat{D} \to D^\star$. $\qquad\square$

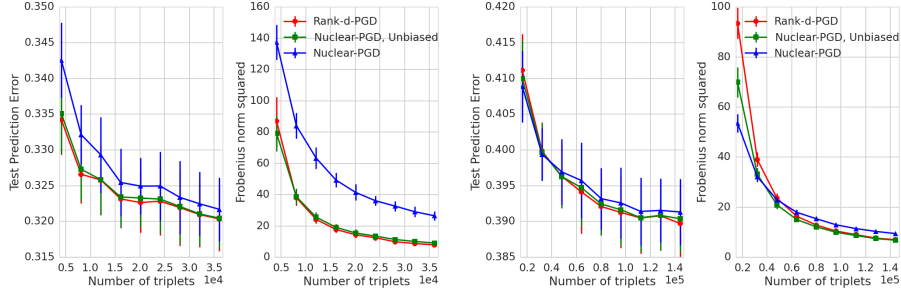

Figure 1: $\boldsymbol{G}^{\star}$ generated with $n = 64$ points in $d = 2$ and $d = 8$ dimensions on the left and right.

## 4 Experimental Study

The section empirically studies the properties of estimators suggested by our theory. It is *not* an attempt to perform an exhaustive empirical evaluation of different embedding techniques; for that see [18, 4, 6, 3]. In what follows each of the $n$ points is generated randomly: $\boldsymbol{x}_i \sim \mathcal{N}(0, \frac{1}{2d}I_d) \in \mathbb{R}^d$, $i = 1, \ldots, n$, motivated by the observation that

$$\mathbb{E}[|\langle \mathcal{L}_t, \boldsymbol{G}^{\star} \rangle|] \;=\; \mathbb{E}\big[\big|\, \|\boldsymbol{x}_i - \boldsymbol{x}_j\|_2^2 - \|\boldsymbol{x}_i - \boldsymbol{x}_k\|_2^2 \,\big|\big] \;\leq\; \mathbb{E}\big[\|\boldsymbol{x}_i - \boldsymbol{x}_j\|_2^2\big] \;=\; 2\mathbb{E}\big[\|\boldsymbol{x}_i\|_2^2\big] \;=\; 1$$

for any triplet $t = (i, j, k)$. We report the prediction error on a holdout set of $10{,}000$ triplets and the error in Frobenius norm of the estimated Gram matrix over 36 random trials. We minimize the logistic MLE objective $\widehat{R}_{\mathcal{S}}(\boldsymbol{G}) = \frac{1}{|S|} \sum_{t \in S} \log(1 + \exp(-y_t \langle \mathcal{L}_t, \boldsymbol{G} \rangle))$.

For each algorithm considered, the domain of the objective variable $\boldsymbol{G}$ is the space of symmetric positive semi-definite matrices. None of the methods impose the constraint $\max_{ij} |G_{ij}| \leq \gamma$ (as done above), since this was used to simplify the analysis and does not have a large impact in practice. *Rank-d Projected Gradient Descent (PGD)* performs gradient descent on the objective $\widehat{R}_{\mathcal{S}}(\boldsymbol{G})$ with line search, projecting onto the subspace spanned by the top $d$ eigenvalues at each step (i.e. setting the smallest $n - d$ eigenvalues to 0). *Nuclear Norm PGD* performs gradient descent on $\widehat{R}_{\mathcal{S}}(\boldsymbol{G})$ projecting onto the nuclear norm ball with radius $\|\boldsymbol{G}^{\star}\|_*$, where $\boldsymbol{G}^{\star}$ is the Gram matrix of the latent embedding. The nuclear norm projection can have the undesirable effect of shrinking the non-zero eigenvalues toward the origin. To compensate for this potential bias, we employ *Nuclear Norm PGD Debiased*, which takes the biased output of *Nuclear Norm PGD*, decomposes it into $\boldsymbol{U}\boldsymbol{E}\boldsymbol{U}^T$ where $\boldsymbol{U} \in \mathbb{R}^{n \times d}$ are the top $d$ eigenvectors, and outputs $\boldsymbol{U}\mathrm{diag}(\widehat{\boldsymbol{s}})\boldsymbol{U}^T$ where $\widehat{s} = \arg\min_{\boldsymbol{s} \in \mathbb{R}^d} \widehat{R}_{\mathcal{S}}(\boldsymbol{U}\mathrm{diag}(\boldsymbol{s})\boldsymbol{U}^T)$. This last algorithm is motivated by the observation that methods for minimizing $\|\cdot\|_1$ or $\|\cdot\|_*$ are good at identifying the true support of a signal, but output biased magnitudes [19]. *Rank-d PGD* and *Nuclear Norm PGD Debiased* are novel ordinal embedding algorithms.

Figure 1 presents how the algorithms behave for $n = 64$ and $d = 2, 8$. We observe that the unbiased nuclear norm solution behaves near-identically to the rank-$d$ solution and remark that this was observed in all of our experiments (see the supplementary materials for other values of $n$, $d$, and scalings of $\boldsymbol{G}^{\star}$). A popular technique for recovering rank $d$ embeddings is to perform (stochastic) gradient descent on $\widehat{R}_{\mathcal{S}}(\boldsymbol{U}^T\boldsymbol{U})$ with objective variable $\boldsymbol{U} \in \mathbb{R}^{n \times d}$ taken as the embedding [18, 4, 6]. In all of our experiments this method produced Gram matrices nearly identical to those produced by our *Rank-d-PGD* method, but *Rank-d-PGD* was an order of magnitude faster in our implementation. Also, in light of our isometry theorem, we can show that the Hessian of $\mathbb{E}[\widehat{R}_{\mathcal{S}}(\boldsymbol{G})]$ is nearly a scaled identity, leading us to hypothesize that a globally optimal linear convergence result for this non-convex optimization may be possible using the techniques of [20, 21]. Finally, we note that previous literature has reported that nuclear norm optimizations like *Nuclear Norm PGD* tend to produce less accurate embeddings than those of non-convex methods [4, 6]. The results imply that *Nuclear Norm PGD Debiased* appears to close the performance gap between the convex and non-convex solutions.

**Acknowledgments** This work was partially supported by the NSF grants CCF-1218189 and IIS-1447449, the NIH grant 1 U54 AI117924-01, the AFOSR grant FA9550-13-1-0138, and by ONR awards N00014-15-1-2620, and N00014-13-1-0129. We would also like to thank Amazon Web Services for providing the computational resources used for running our simulations.

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
