[Supplementary Material]

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

# 5 Supplementary Materials for "Finite Sample Error Bounds for Ordinal Embedding"

## 5.1 Proof of Lemma 1

**Lemma 1.** *For all $t \in \mathcal{T}$,*

$$\lambda_1(\mathcal{L}_t) = \|\mathcal{L}_t\| = \sqrt{3}$$

*in addition if $n \geq 3$*

$$\|\mathbb{E}_t[\mathcal{L}_t^2]\| = \frac{6}{n-1} \leq \frac{9}{n}$$

*Proof.* Note that $\mathcal{L}_t^3 - 3\mathcal{L}_t = 0$ for all $t \in \mathcal{T}$. Thus by the Cayley-Hamilton theorem, $\sqrt{3}$ is the largest eigenvalue of $\mathcal{L}_t$. A computation shows that the submatrix of $\mathcal{L}_t^2$ corresponding to $i, j, k$ is

$$\begin{pmatrix} 2 & -1 & -1 \\ -1 & 2 & -1 \\ -1 & -1 & 2 \end{pmatrix}$$

and every other element of $\mathcal{L}_t^2$ is zero. Summing over the $t \in \mathcal{T}$ then gives,

$$\mathbb{E}[\mathcal{L}_t^2] = \frac{1}{n\binom{n-1}{2}} \sum_{t \in \mathcal{T}} \mathcal{L}_t^2 = \begin{pmatrix} \frac{6}{n} & \frac{-6}{n(n-1)} & \cdots & \frac{-6}{n(n-1)} \\ \frac{-6}{n(n-1)} & \frac{6}{n} & \cdots & \frac{-6}{n(n-1)} \\ \vdots & \cdots & \cdots & \vdots \\ \frac{-6}{n(n-1)} & \cdots & \frac{-6}{n(n-1)} & \frac{6}{n} \end{pmatrix}$$

This matrix can be rewritten as $\frac{6}{n} I - \frac{6}{n(n-1)} J$. The eigenvalues of $J$ are $-1$ with multiplicity $n-1$ and $n-1$ with multiplicity 1. Hence the largest eigenvalue of $\mathbb{E}[\mathcal{L}_t^2]$ is $\frac{6}{n-1}$. $\qquad\square$

## 5.2 Proof of Theorem 2

*Proof.* For $y, z \in (0,1)$ let $g(z) = z \log \frac{z}{y} + (1-z) \log \frac{1-z}{1-y}$. Then $g'(z) = \log \frac{z}{1-z} - \log \frac{y}{1-y}$ and $g''(z) = \frac{1}{z(1-z)}$. By taking a Taylor series around $y$,

$$g(z) \geq \frac{(z-y)^2/2}{\sup_{x \in [0,1]} x(1-x)} \geq 2(z-y)^2.$$

Now applying this to $z = f(\langle \mathcal{L}_t, \boldsymbol{G}^\star \rangle)$ and $y = f(\langle \mathcal{L}_t, \boldsymbol{G} \rangle)$ gives

$$f(\langle \mathcal{L}_t, \boldsymbol{G}^\star \rangle) \log \frac{f(\langle \mathcal{L}_t, \boldsymbol{G}^\star \rangle)}{f(\langle \mathcal{L}_t, \boldsymbol{G} \rangle)} + (1 - f(\langle \mathcal{L}_t, \boldsymbol{G}^\star \rangle)) \log \frac{1 - f(\langle \mathcal{L}_t, \boldsymbol{G}^\star \rangle)}{1 - f(\langle \mathcal{L}_t, \boldsymbol{G} \rangle)} \geq 2(f(\langle \mathcal{L}_t, \boldsymbol{G}^\star \rangle) - f(\langle \mathcal{L}_t, \boldsymbol{G} \rangle))^2$$
$$\geq 2C_f^2 (\langle \mathcal{L}_t, \boldsymbol{G}^\star \rangle - \langle \mathcal{L}_t, \boldsymbol{G} \rangle)^2$$

where the last line comes from applying Taylor's theorem to $f$, $f(x) - f(y) \geq \inf_{z \in [x,y]} f'(z)(x-y)$ for any $x, y$. Thus

$$\begin{aligned} R(\boldsymbol{G}) - R(\boldsymbol{G}^\star) &= \frac{1}{|\mathcal{T}|} \sum_{t \in \mathcal{T}} f(\langle \mathcal{L}_t, \boldsymbol{G}^\star \rangle) \log \frac{f(\langle \mathcal{L}_t, \boldsymbol{G}^\star \rangle)}{f(\langle \mathcal{L}_t, \boldsymbol{G} \rangle)} + (1 - f(\langle \mathcal{L}_t, \boldsymbol{G}^\star \rangle)) \log \frac{1 - f(\langle \mathcal{L}_t, \boldsymbol{G}^\star \rangle)}{1 - f(\langle \mathcal{L}_t, \boldsymbol{G} \rangle)} \\ &\geq \frac{2C_f^2}{|\mathcal{T}|} \sum_{t \in T} (\langle \mathcal{L}_t, \boldsymbol{G}^\star \rangle - \langle \mathcal{L}_t, \boldsymbol{G} \rangle)^2 \\ &= \frac{2C_f^2}{|\mathcal{T}|} \sum_{t \in T} (\langle \mathcal{L}_t, \boldsymbol{G} - \boldsymbol{G}^\star \rangle)^2 = \frac{2C_f^2}{|\mathcal{T}|} \|\mathcal{L}(\boldsymbol{G}) - \mathcal{L}(\boldsymbol{G}^*)\|_2^2. \end{aligned}$$

$$\square$$

## 5.3 Proof of Lemma 3

**Lemma 3.** *Let $D$ and $D'$ be two different distance matrices of $n$ points in $\mathbb{R}^d$ and $\mathbb{R}^{d'}$ respectively. Let $C$ and $C'$ be the components of $D$ and $D'$ orthogonal to $J$. Then*

$$n\|C - C'\|_F^2 \;\leq\; \|\Delta(C) - \Delta(C')\|^2 = \|\Delta(D) - \Delta(D')\|^2 \;\leq\; 2(n-1)\|C - C'\|_F^2 \; .$$

We can view the operator $\Delta$ defined above as acting on the space $\mathbb{R}^{\binom{n}{2}}$ where each symmetric hollow matrix is identified with vectorization of it's upper triangular component. With respect to this basis $\Delta$ is an $n\binom{n-1}{2} \times \binom{n}{2}$ matrix, which we will denote by $\Delta$. Since $C$ and $C'$ are orthogonal to the kernel of $\Delta$, the lemma follows immediately from the following characterization of the eigenvalues of $\Delta^T\Delta$.

**Lemma 6.** $\Delta^T\Delta : \mathbb{S}_h^n \to \mathbb{S}_h^n$ *has the following eigenvalues and eigenspaces,*

- *Eigenvalue $0$, with a one dimensional eigenspace.*

- *Eigenvalue $n$, with a $n - 1$ dimensional eigenspace.*

- *Eigenvalue $2(n - 1)$, with a $\binom{n}{2} - n$ dimensional eigenspace.*

*Proof.* The rows of $\Delta$ are indexed by triplets $t \in \mathcal{T}$ and columns indexed by pairs $i, j$ with $1 \leq i < j \leq n$ and vice-versa for $\Delta^T$. The row of $\Delta^T$ corresponding to the pair $i, j$ is supported on columns corresponding to triplets $t = (l, m, n)$ where $m < n$ and $l$ and one of $m$ or $n$ form the pair $i, j$ or $j, i$. Specifically, letting $[\Delta^T]_{(i,j),t}$ denote the entry of $\Delta^T$ corresponding to row $i, j$ and column $t$,

- if $l = i, m = j$ then $[\Delta^T]_{(i,j),t} = 1$

- if $l = i, n = j$ then $[\Delta^T]_{(i,j),t} = -1$

- if $l = j, m = i$ then $[\Delta^T]_{(i,j),t} = 1$

- if $l = j, n = i$ then $[\Delta^T]_{(i,j),t} = -1$

Using this one can easily check that

$$[\Delta^T\Delta D]_{i,j} = \sum_{(i,j,k)\in\mathcal{T}} D_{ij} - D_{ik} - \sum_{(i,k,j)\in\mathcal{T}} D_{ik} - D_{ij} + \sum_{(j,i,k)\in\mathcal{T}} D_{ji} - D_{jk} - \sum_{(j,k,i)\in\mathcal{T}} D_{jk} - D_{ji}$$

$$= 2(n-1)D_{ij} - \sum_{n\neq i} D_{in} - \sum_{n\neq j} D_{jn}. \tag{11}$$

This representation allows us to find the eigenspaces mentioned above very quickly.

**Eigenvalue** $0$. From the above discussion, we know the kernel is generated by $J = \mathbf{1}\mathbf{1}^T - I$.

**Eigenvalue** $2(n - 1)$. This eigenspace corresponds to all symmetric hollow matrices such that $D\mathbf{1} = \mathbf{0}$. For such a matrix each row and column sum is zero and so in particular, the sums in (11) are both zero. Hence for such a $D$,

$$[\Delta^T\Delta D]_{i,j} = 2(n-1)D_{ij}$$

The dimension of this subspace is $\binom{n}{2} - n$, indeed there are $\binom{n}{2}$ degree of freedom to choose the elements of $D$ and $D\mathbf{1} = \mathbf{0}$ adds $n$ constraints.

**Eigenvalue** $n$. This eigenspace corresponds to the span of the matrices $D^{(i)}$ defined as,

$$D^{(i)} = -n(e_i\mathbf{1}^T + \mathbf{1}e_i^T - 2e_ie_i^T) + 2J$$

where $e_i$ is the standard basis vector with a $1$ in the $i$th row and $0$ elsewhere. As an example,

$$D^{(1)} = \begin{pmatrix} 0 & -n+2 & \cdots & -n+2 \\ \vdots & 2 & \cdots & 2 \\ -n+2 & 2 & \cdots & 0 \end{pmatrix}.$$

If $i, j \neq m$, then $D_{ij}^{(m)} := [\boldsymbol{D}^{(m)}]_{ij} = 2$, and we can compute the row and column sums

$$\sum_{n \neq i} D_{in}^{(m)} = \sum_{n \neq j} D_{jn}^{(m)} = 2(n-2) - n + 2 = n - 2.$$

This implies that $D_{ij}^{(m)} = n$, and so by (11)

$$[\Delta^T \Delta \boldsymbol{D}^{(m)}]_{i,j} = 2(n-1) \cdot 2 - (n-2) - (n-2) \;=\; 2n \;=\; n D_{ij}^{(m)}.$$

Otherwise, without loss of generality we can assume that $i = m, j \neq m$ in which case, $[\boldsymbol{D}^{(m)}]_{ij} = -n + 2$, the row and columns sums can be computed as

$$\sum_{n \neq i} D_{in}^{(m)} = (n-1)(-n+2)$$

and

$$\sum_{n \neq j} D_{in}^{(m)} = n - 2.$$

Putting it all together,

$$\begin{aligned}
[\Delta^T \Delta \boldsymbol{D}^{(m)}]_{m,j} &= 2(n-1) \cdot (-n+2) - (n-1)(-n+2) - (n-2) \\
&= (n-1)(-n+2) + (-n+2) \\
&= n(-n+2) \\
&= n D_{m,j}^{(m)}
\end{aligned}$$

and $\Delta^T \Delta \boldsymbol{D} = n\boldsymbol{D}$. Note that the dimension of $\operatorname{span}\langle \boldsymbol{D}^{(i)} \rangle = n - 1$ since

$$\sum_m \boldsymbol{D}^{(m)} = 0.$$

$\square$

### 5.4  Proof of Lemmas 4

**Lemma 4.** *Let $\boldsymbol{D}$ be a Euclidean distance matrix on $n$ points. Then $\boldsymbol{D}$ is negative semidefinite on the subspace*

$$\mathbf{1}^\perp := \{ \boldsymbol{x} \in \mathbb{R}^n | \mathbf{1}^T \boldsymbol{x} = 0 \}.$$

*Furthermore,* $\ker(\boldsymbol{D}) \subset \mathbf{1}^\perp$.

*Proof.* The associated Gram matrix $\boldsymbol{G} = -\frac{1}{2} \boldsymbol{V} \boldsymbol{D} \boldsymbol{V}$ is a positive semidefinite matrix. For $\boldsymbol{x} \in \mathbf{1}^\perp$, $\boldsymbol{J}\boldsymbol{x} = -\boldsymbol{x}$ so

$$\boldsymbol{x}^T \left( -\frac{1}{2} \boldsymbol{V} \boldsymbol{D} \boldsymbol{V} \right) \boldsymbol{x} = -\frac{1}{2} \boldsymbol{x}^T \boldsymbol{D} \boldsymbol{x} \leq 0$$

establishing the first part of the theorem. Now if $\boldsymbol{x} \in \ker \boldsymbol{D}$,

$$0 \leq -\frac{1}{2} \boldsymbol{x}^T \boldsymbol{V} \boldsymbol{D} \boldsymbol{V} \boldsymbol{x} = -\frac{1}{2} \boldsymbol{x}^T \mathbf{1}\mathbf{1}^T \boldsymbol{D} \mathbf{1}\mathbf{1}^T \boldsymbol{x} = -\frac{1}{2} \mathbf{1}^T \boldsymbol{D} \mathbf{1} (\mathbf{1}^T \boldsymbol{x})^2 \leq 0,$$

where the last inequality follows from the fact that $\mathbf{1}^T \boldsymbol{D} \mathbf{1} > 0$ since $\boldsymbol{D}$ is non-negative. Hence $\mathbf{1}^T \boldsymbol{x} = 0$ and $\ker \boldsymbol{D} \subset \mathbf{1}^\perp$. $\square$

## 6  Additional Empirical Results

Figure 2: **Varying dimension** $n = 64$, $d = \{1, 2, 4, 8\}$ from top to bottom.

Figure 3: **Varying Noise** $n = 64$, $d = 2$, $\alpha M$ for $\alpha = \{2, 1, \frac{1}{2}, \frac{1}{4}\}$ from top to bottom.

Figure 4: **Varying # items** $n = \{16, 32, 64, 128\}$, $d = 2$ from top to bottom.