[Reviews · NeurIPS 2016]

Reviewer 1

Summary

This paper deals with ordinal embedding of items. Given a set of (noisy) distance comparisons in the form of triplets ("item i is closer to item j than to item k"), the goal is to predict well unseen triplets and ideally to recover the true distance matrix of the latent embedding of the items. The authors derive a finite sample prediction error bound for a regularized empirical risk minimization formulation of the problem, and when the noise process is known, approximation bounds for the estimated (centered) distance matrix, and an exact recovery result when the number of observed triplets goes to infinity. As a side contribution, the authors also propose some new embedding algorithms.

Qualitative Assessment

I enjoyed reading this submission quite a lot. The problem and notations are carefully introduced. The authors then elegantly formulate noisy ordinal embedding as a (constrained) regularized risk minimization problem (somewhat similar to kernel learning), and derive several interesting and nontrivial results for the empirical risk minimizer. I am not an expert in ordinal embedding but the results seem quite novel and insightful. However I am not completely convinced of the validity of Theorem 1, which most subsequent results are based on. In the proof, the authors rely on some statistical learning theory techniques which require that the empirical risk is a sum of *independent* terms. However, although the triplets in \mathcal{S} are sampled uniformly and independently with replacement from \mathcal{T}, it seems that any two triplets in \mathcal{S} with at least one overlapping index are not independent since both labels depend on some common distance entries. Please clarify if I'm mistaken. Otherwise, Theorem 1 could maybe be fixed by relying on results from ERM with (incomplete) U-statistics, see [1]. Some other comments: - For the results of Sections 3 and 4, it seems that one needs to assume that the link function f is differentiable. - To avoid the projection onto the nuclear norm ball (which requires eigenvalue decomposition of G and is thus expensive when n is large), one could use a Frank-Wolfe algorithm [2], whose iterative rank-1 updates only require the computation of the leading eigenvector of the gradient of G. - Could the Nuclear Norm PGD benefit from performing the debiasing step at each iteration? Or even to perform fully corrective updates in the same subspace? This has been shown to help for coordinate descent algorithms (very similar to Frank-Wolfe) on nuclear norm problems (see for instance [3]). Typos: - Equation after line 39: the first x_k should be x_j. - Line 51: double "a" before "non-empty". - Lines 70-71: X and G should be in bold. - In Theorem 2, I think \mathbf{G} should be in \mathcal{G}_{\lambda,\gamma}. - Line 230: remove "fact" or "observation" [1] Clémençon et al. Scaling-up Empirical Risk Minimization: Optimization of Incomplete U-statistics. JMLR 2016. [2] Jaggi. Revisiting Frank-Wolfe: Projection-Free Sparse Convex Optimization. ICML 2013. [3] Blondel et al. Convex Factorization Machines. ECML/PKDD 2015.

Confidence in this Review

2-Confident (read it all; understood it all reasonably well)


Reviewer 2

Summary

The authors derive PAC-style finite sample bounds for ordinal embedding from noisy triplet data. Furthermore they relate prediction error with embedding quality for known noise models and propose practical algorithms for recovering the embedding.

Qualitative Assessment

I really like that the authors have formulated and analyzed a clean framework for ordinal embedding that can account for various noise models. The theoretical results and proofs are stated cleanly and I believe that this work would be a nice addition to the current theory of ordinal embeddings. Minor comments: - It would be instructive for the reader if ‘centered distance matrix’ (Section 4, line 142) is briefly explained. - some typos in lines 30, 51, 214.

Confidence in this Review

3-Expert (read the paper in detail, know the area, quite certain of my opinion)


Reviewer 3

Summary

This paper addresses the problem of finding potentially low dimensional embedding from distance comparisons known as ordinal embedding. The authors provide novel generalization bounds based on finite quantitiesfor ordinal embedding, as far as I understand the fiel, this is novel. They also study a particular model based on maximum likelihood estimation where they justify that a correct embedding can be recevered. An experimental evaluation is provided on synthetic data that includes 2 new algorithms.

Qualitative Assessment

Pros: -Interesting and a priori novel theoretical analysis for Ordinal Embedding Cons: -Paper sometimes hard to follow where some parts deserve more attention -The noise model the authors propose to study is not clear -Experimental evaluation limited to synthetic data with comparison to only one existing method Overall: Good and interesting theoretical results that are novel for this topic. But the paper lacks of discussion/precision on some parts and the practical aspects are relatively limited. Comments: -The bounds proposed in this paper appear to be novel and are interesting. I wonder wether there exists a link with U-statistics that can be used to justify that a true risk can be efficiently estimated by considering a random subset from all the possible triplets. Check the following reference for example: *S. Clemencon, I. Colin and A. Bellet. Scaling-up Empirical Risk Minimization: Optimization of Incomplete U-statistics. Journal of Machine Learning Research 17(76):1-36, 2016. The results in Section 4 are interesting, but the authors do not discuss their implication from a practical standpoint. We do not know if they are purely theoretical to justify the consistency of the method with no ambition to deduce some practical algorithms or if there are some perspectives to develop new well-founded approach to recover the embedding. -The authors mention the presence of a noise model but I was unable to find a description or discussion about this noise model in the paper. If the authors have in mind a classic Gaussian noise, this must be clearly mentioned and discussed accordingly in the paper. In the supplementary caption of Figure 3 page 14, I was not able to understand what means \alphaM for describing the noise since \alpha and M have not been introduced before. ==After rebuttal == Ok for the answer. You may anyway specify more clearly your point here. Since $f$ takes as argument the distance between elements in the triplet (which is reasonable for defining the true labeling), one could think that the noise may depend on this distance which is not what you meant here. -The experimental setup is limited to synthetic data but i would have been interesting to study the behavior of the approach on real data. The notion of prediction error is not explicitly mention in the setup, I guess that this corresponds to the evaluation over the test set of the difference between the risk R in Section 2 of the learned model and the true R defined. If so, I think that an additional experiment showing the accuracy obtained on the test set by counting the number of times the distances between in each triplet are correctly ordered could be interesting to assess if the embedding is relevant in addition to a good estimation of the true matrix G^*. I am not completely convinced by the approach "Nuclear Norm PGD Debiased" since it exhibits a performance similar or worse than "Nuclear PGD". Anyway the two proposed algorithms are presented quickly is Section 5. However, they deserve more discussion. In particular, the remark on performance gap between convex and non convex solutions at the end of Section 5. The algorithms are not sufficiently presented and their properties are not sufficiently discussed. I think that the authors could move a part of the proofs of justifications of Section 4 into the supplementary material to have more place to discuss other aspects of the paper. Finally, the experimental section seems disconnected from the results presented in Section 4. Trying to assess in practice the quantities presented in this section from the model obtained in the experiments could strengthen the global coherence of the paper. -The paper lacks of a general conclusion that could present the perspective and replace the work with respect to existing work in the state of the art. At the end, it is difficult for me to really assess the quality of the contribution in comparison to existing work in the literature and I think the paper misses a deeper discussion on existing work. -Other remarks: *l51: has a a non-empty -> has a non-empty *l77: in the definition of all the triplets, T, you may need to add the constraint: i < j *l89: If you assume that the set of triplets T is infinite, the notation \frac{1}{T} is not accurate in the definition of the expectation *l98: later we will link define $\ell$ -> choose between link and define

Confidence in this Review

2-Confident (read it all; understood it all reasonably well)


Reviewer 4

Summary

This paper provides a non-asymptotic bound for the estimation of inner product matrix (G) from ordinal information. A risk is provided from a probabilistic model, and the estimated G is not much far from the minimal achievable risk with high probability.

Qualitative Assessment

I understand the importance of the problem of embedding using the ordinal information but do not fully catch up the current research on this problem. I briefly claim one concern in the provided theorems: The definition of G* is not clear. Is G* the true inner product matrix or G* with minimum R(G)? With infinite data, probably they are the same. But with finite n number of data, the two can be different: the G with minimum R(G) for all possible triplets of n data is different from the true G from the same n number of data.

Confidence in this Review

1-Less confident (might not have understood significant parts)